# The Genomic Topography of Appendiceal Cancers: Our Current Understanding, Clinical Perspectives, and Future Directions

**DOI:** 10.3390/cancers17193275

**Published:** 2025-10-09

**Authors:** Daniel J. Gironda, Richard A. Erali, Steven D. Forsythe, Ashok K. Pullikuth, Rui Zheng-Pywell, Kathleen A. Cummins, Shay Soker, Xianyong Gui, Edward A. Levine, Konstantinos I. Votanopoulos, Lance D. Miller

**Affiliations:** 1Department of Cancer Biology, Wake Forest University School of Medicine, Winston-Salem, NC 27157, USA; daniel.gironda@wfusm.edu (D.J.G.); ashok.pullikuth@advocatehealth.org (A.K.P.); 2Wake Forest Organoid Research Center, Wake Forest Institute for Regenerative Medicine, Wake Forest University School of Medicine, Winston-Salem, NC 27101, USA; 3Department of Surgery, Section of Surgical Oncology, Atrium Health Wake Forest Baptist, Winston-Salem, NC 27157, USA; rui.zhengpywell@advocatehealth.org (R.Z.-P.);; 4Neuroendocrine Cancer Therapy Section, National Cancer Institute, National Institutes of Health, Bethesda, MD 20892, USA; 5Wake Forest Institute for Regenerative Medicine, Wake Forest University School of Medicine, Winston-Salem, NC 27101, USA; 6Atrium Health Wake Forest Baptist Comprehensive Cancer Center, Winston-Salem, NC 27157, USA; 7Department of Pathology, Wake Forest University School of Medicine, Winston-Salem, NC 27157, USA

**Keywords:** appendix cancer, colon cancer, bowel malignancy, molecular profiling, next-generation sequencing, mutational landscape, precision medicine

## Abstract

**Simple Summary:**

Appendiceal cancer (AC) is a rare and often fatal malignancy with poorly understood pathogenesis and limited treatment options. Although recent genomic studies have begun to shed light on the more prominent genetic drivers of AC, the rarity of the disease has impeded the comprehensive genomic characterization of its diverse histologic subtypes. In this review, we synthesize the current literature on the genomic landscape of AC, highlighting subtype-specific mutational profiles, their prognostic implications, and the emerging role of molecular profiling in guiding future research and shaping prospective therapeutic strategies.

**Abstract:**

**Background/Objectives**: Appendiceal cancer (AC) is a rare and understudied malignancy with limited genomic data available to guide clinical interventions. Historically treated as a subtype of colorectal cancer, AC is now recognized as a distinct disease with unique histologic subtypes and molecular features. This review aims to consolidate current genomic data across AC subtypes and explore the clinical relevance of recurrent mutations. **Methods**: A systematic literature review was performed in accordance with general Preferred Reporting Items for Systemic Reviews and Meta-Analyses (PRISMA) guidelines. Using search engines such as PubMed and Web of Science, we selected studies based on relevance to AC genomics using search terms such as “appendix cancer”, “appendiceal cancer”, “pseudomyxoma peritonei”, “sequencing”, “mutation”, and “genotype”. **Results**: AC comprises five major histologic subtypes—appendiceal neuroendocrine neoplasms (ANENs), mucinous appendiceal neoplasms (MANs), goblet cell adenocarcinomas (GCAs), colonic-type adenocarcinomas (CTAs) and signet ring cell adenocarcinomas (SRCs)—each with unique clinical behaviors and mutational profiles. Low-grade tumors, such as ANENs and MANs, frequently harbor *KRAS* and *GNAS* mutations, while high-grade subtypes, such as CTAs and SRCs, are enriched for *TP53*, *APC*, and *SMAD* gene alterations. GCA tumors exhibit a distinct mutational spectrum involving chromatin remodeling genes such as *ARID1A* and *KMT2D*. Compared to colorectal cancer, AC demonstrates lower frequencies of *APC* and *TP53* mutations and a higher prevalence of *GNAS* mutations, consistent with a pathological divergence from CRC. **Conclusions**: The genomic heterogeneity of AC is commensurate with its histological complexity and has important implications for diagnosis, prognosis and treatment. While certain actionable mutations are present in a subset of tumors, large-scale genomic characterization efforts and development of subtype-specific models will be essential for advancing precision medicine in AC.

## 1. Introduction

Appendiceal cancer (AC) is a rare malignancy of the gastrointestinal tract associated with variable patterns of malignant spread and patient survival [1,2]. The estimated incidence of AC in the United States is approximately 6 cases per 1,000,000 individuals annually, with an increasing trend observed over the past two decades [3,4,5,6]. Clinical outcomes are largely influenced by tumor grade, tumor burden at diagnosis, histologic type, medical comorbidities, performance status (i.e., Eastern Cooperative Oncology Group and Karnofsky), and completeness of surgical cytoreduction, with AC 5-year survival ranging from 18% in advanced disease to 97% in well-differentiated neuroendocrine tumors [6,7,8,9,10,11,12].

To date, no screening programs are used clinically for the early detection of AC. AC is incidentally diagnosed in roughly 1% of appendectomies, with most tumors being detected after clinical presentation of acute appendicitis secondary to tumoral obstruction of the appendiceal lumen [13,14,15,16,17]. Given that the appendix embryonically arises from the cecum [18] and remains as an anatomical constituent of the colon throughout development, AC has historically been categorized and treated as a variant of colorectal cancer (CRC). However, AC remains histologically distinct from CRC and is resistant to most standard CRC treatment regimens [4,19]. At present, the therapeutic management of AC depends largely on tumor grade and surgical resectability. Non-resectable cases are treated with palliative chemotherapy [13,20], whereas resectable tumors are often managed with cytoreductive surgery (CRS), followed by hyperthermic intraperitoneal chemotherapy (HIPEC) and, in some cases, adjuvant chemotherapy [10,13,20,21].

Histological classification of AC has evolved over the past few decades, with the American Joint Committee on Cancer (AJCC) 8th Edition now recognizing five major subtypes: appendiceal neuroendocrine neoplasms (ANENs), mucinous appendiceal neoplasms (MANs—which includes non-infiltrating low-grade appendiceal mucinous neoplasms (LAMNs) and high-grade appendiceal mucinous neoplasms (HAMNs), and invasive mucinous adenocarcinomas (MAAs), goblet cell adenocarcinomas (GCAs), colonic-type adenocarcinomas (CTAs), and signet ring cell adenocarcinomas (SRCs) [3,22,23,24] (Figure 1 and Figure 2a). These subtypes differ significantly in their clinical behavior. For instance, poorly differentiated tumors and those with signet ring morphology portend a worse prognosis, while ANENs and well-differentiated MANs often follow a more indolent disease course (Figure 2b).

In AC, two main grading systems are used clinically: the three-tiered AJCC system (G1–G3) based on cellular differentiation, and the dichotomous Chicago Consensus system that categorizes tumors into low- and high-grade disease [24,25,26]. Grading involves the pathologic assessment of cellular features that include nuclear architecture, tumor cellularity, mitotic frequency, and the presence of signet ring morphology [24]. Low-grade appendiceal tumors can present as LAMNs and HAMNs, or invasive low-grade MAAs that present as well-differentiated cells with dysmorphic cellularity of the superficial epithelia [27]. High-grade AC is characterized by moderate to poor cellular differentiation and tissue invasion. These tumors are clinically more aggressive and include moderately to poorly differentiated ANENs, MANs, CTAs, GCAs, as well as SRCs [26]. A detailed schema of the relationship between histologic subtype and grade is shown in Figure 1.

Peritoneal carcinomatosis (PC) frequently arises in AC as a result of appendix rupture and the subsequent dissemination of tumor cells throughout the peritoneum [28]. In this metastatic state, the clinical aggressiveness of AC is largely dependent on tumor grade—patients with low-grade disease exhibit a 5-year survival of 84%, compared to just 48% in those with high-grade tumors [29].

Although our understanding of the mutational drivers of AC is still emerging, genomic studies have begun to identify patterns of somatic mutations linked to specific histologic subtypes (Figure 2b; Table 1). Most studies to date have been derived from small cohorts and targeted sequencing efforts using a variety of diverse gene panels including the MSK-IMPACT, Ion Ampliseq Cancer Panel, Oncomine Comprehensive Cancer Panel, and Ion AmpliSeq Comprehensive Cancer panel, which vary widely with respect to the breadth and type of genes represented. Despite preliminary successes in the targeted sequencing era, the need for broader characterization approaches like whole-exome and whole-genome sequencing remains for AC patients. This review consolidates recent genomic findings, compares mutational frequencies and their clinical implications across AC subtypes, and highlights opportunities for translational research.

## 2. Appendiceal Neuroendocrine Neoplasms

Gastroenteropancreatic neuroendocrine neoplasms (GEP-NENs) arise from neuroendocrine cell types including enterochromaffin cells, L-cells, as well as tubular epithelial cells, and are histologically defined by well-differentiated growth patterns. Among these tumors, the appendix is the fifth most common site of origin [41,42]. Appendiceal NENs (ANENs) are the most frequently occurring appendiceal cancer histologic subtype [43] (Figure 2a).

ANENs are generally associated with favorable clinical outcomes. Surveillance, Epidemiology, and End Results (SEER) data indicate a 5-year overall survival (OS) of approximately 90%, with some studies reporting rates as high as 97% [44,45,46,47,48,49,50] (Figure 2b). Grading of appendiceal NENs is typically based on the World Health Organization (WHO) and AJCC classification schemes, which stratify tumors as well-differentiated (NEN-G1; KI-67 expression < 3%), moderately differentiated (NEN-G2; KI-67 = 3–20%), and poorly differentiated (NEN-G3; KI-67 > 20%), with a separate category for mixed adenoneuroendocrine carcinomas (MANECs) [50,51,52,53]. Unlike other appendiceal malignancies, ANENs generally exhibit a low overall burden of somatic mutations and chromosomal aberrations. This has led to the hypothesis that ANEN pathogenesis is more dependent on tumor grade and differentiation status rather than widespread genomic instability [8,54].

The most frequently mutated genes in ANENs are *TP53* (average 23.1%, range: 10.7–44%), *APC* (11.4%, 3.3–28.6%), *SMAD4* (11.5%, 0–40%), and *KRAS* (11.4%, 0–28.6%), as extrapolated from multiple reports [30,35,36,40,55] (Figure 3a; Appendix A). While these alterations are commonly seen in colorectal and other appendiceal cancer subtypes, their presence in ANENs may reflect tumor heterogeneity or sampling from mixed phenotypes. A broader analysis of 84 primary GEP-NENs, including only five appendiceal NENs, identified *TP53* (13.1%), *PTEN* (7.1%), *SMAD4* (6%), *EGFR* (6%), *ATM* (4.8%), *CDKN2A* (6%), and *KIT* (4.8%) as the most commonly altered genes [30]. These data suggest that *TP53* and *SMAD4* mutations may be shared among NENs across the GI tract, whereas *KRAS* and *APC* mutations may be more specific to the appendiceal subtype.

Germline and somatic mutations in several genes have been implicated in GEP-NEN development [8,54]. For instance, *CDKN1B*, which encodes a cyclin-dependent kinase inhibitor, has been identified as a tumor suppressor gene whose inactivation promotes unchecked cell cycle progression [56]. In addition, genes such as *IPMK*, *MEN1*, *VHL*, *TSC2* and *NF* have been identified as potential germline drivers of intestinal NEN [7,57,58], although their precise roles in appendiceal NEN remain unclear.

Among ileal carcinoids, hemizygous deletion of chromosome 18q is the most frequent chromosomal aberration and is present in 61–74% of cases [8,59]. While similar deletions have been observed in appendiceal NENs, robust evidence is lacking. Genomic susceptibility to pancreatic NENs has been linked to inherited mutations in *MEN1*, *DAXX*, *ATRX*, *VHL*, *NF1*, *EPAS1*, *TSC1*, and *CDKN1B*, with potential relevance to duodenal and gastric NENs as well [8,54]. However, the frequency and impact of these mutations in appendiceal NENs remain largely unknown.

## 3. Mucinous Neoplasms of the Appendix

Mucinous appendiceal neoplasms (MANs) are known to cause malignant pseudomyxoma peritonei (PMP)—a condition characterized by the excessive production of mucin by neoplastic epithelial cells, resulting in mucin accumulation within the peritoneal cavity [60,61]. In these tumors, mucin may account for over 50% of the acellular tumor volume [4,62,63]. Mucinous appendiceal adenocarcinoma (MAA) is the predominant invasive form of MAN, and a recent analysis of the SEER database encompassing 13,456 patients with pathologically confirmed MAA reported a 5-year mortality rate of 55% [44]. One- and two-year survival rates were 90.5% and 78.5%, respectively [6,31,45,64,65,66,67,68,69,70] (Figure 2b).

The most prevalent genetic alterations observed in MAAs are missense mutations in *KRAS*, *GNAS* and *TP53*, with estimated frequencies of 67% (range: 44–100%), 44% (28–69%), and 28% (10–40%), respectively [9,12,32,33,34,35,36,38,39,40,71,72,73,74] (Figure 3b). Notably, the mutational frequencies of *KRAS* and *GNAS* appear to correlate with tumor grade. In a study analyzing MAA tumors stratified by the AJCC grading system, KRAS mutations were present in 61% of G1, 72% of G2, and only 19% of G3 tumors [25]. Analysis of open-source data on 164 MAA tumors generated by an MSK-Impact Trial that followed the WHO classification system [38] showed a similar grade-dependent trend with KRAS mutations rates of 88% (G1), 83% (G2), and 53% (G3)—the higher overall frequencies possibly owing to variation in grade classification system or sequencing method. When classified using the Chicago Consensus’ grading system, *KRAS* alterations were found in 60% of low-grade and 56% of high-grade PMP cases [74]. In a similar vein, *GNAS* mutations are more prevalent in low-grade MAAs (~70%) as compared to high-grade MAAs (~17%) [3,12,34,75], suggesting a potential role in early tumorigenesis. Furthermore, early-onset MAAs (<50 years) were found to have significantly lower odds of harboring *GNAS* mutations as compared to late-onset cases (19% vs. 29%) [76]. These observations suggest that *RAS* and *GNAS* pathway activation are key oncogenic events in the development of low-grade MAAs in older patients, whereas alternative pathways may predominantly drive early onset and high-grade disease [25,77,78].

*TP53* mutations, which are common (~70%) in colorectal cancer (CRC), occur in only ~28% of MAAs overall but are more frequent in high-grade tumors (47%) than low-grade tumors (32%) [72,74]. Inactivating mutations in *SMAD4*, a critical mediator of TGF-β signaling, was found in 24% of high-grade MAAs (G2/G3), and in 0% of low-grade (G1) tumors, further implicating TGF-β pathway deregulation in high-grade disease [79].

Additional genomic alterations have been reported in MAAs including *COX2* expression (61%) and mutations in *PIK3CA* (17%), *SMAD2* (20%), *SMAD3* (20%), *RNF43* (15%), *TGFBR1* (10%) and *TGFBR2* (10%) [3,9,32,35,38,39,40,70]. High-grade MAAs are more likely to harbor alterations in *RNF43*, *APC*, *TP53*, *SMAD2*, and *SMAD3*, thus highlighting potential molecular distinctions between low- and high-grade disease.

Loss of heterozygosity (LOH)—the loss of one or more functional copies of a particular gene—was found to be a relatively rare phenomenon in MAAs as 76% of cases demonstrated a low fractional mutation rate [79]. However, a high fractional mutation rate, i.e., the occurrence of ≥25% mutations at a particular chromosomal locus, was found to occur at chromosomes 18q and 17p in MAAs and with greater frequency in G2 (31%) and G3 (38%) disease as compared to G1 (14%). No grade-related LOH differences were observed at chromosomes 1p, 3p, 5q, 7q, 9q, 9p, 10q [79]. Microsatellite instability (MSI) is a measure of DNA mutation frequency owing to loss of DNA damage repair mechanisms. While associated with pathogenicity in 15% of CRCs, MSI appears to be an infrequent phenomenon (0.9%) among MAAs [5,9,80].

## 4. Appendiceal Goblet Cell Adenocarcinoma

Goblet cell adenocarcinomas (GCAs) represent approximately 14–19% of primary appendiceal cancers [6,81] (Figure 2a). These tumors exhibit pleiomorphic histologic features, with elements of both epithelial and neuroendocrine differentiation, often arising from goblet cells and infrequently incorporating Paneth cells [74,82,83]. Historically, GCAs have been referred to by several terms, including “mucinous carcinoid,” “mixed crypt cell carcinoma,” and “adenocarcinoid”, which underscores their histologic complexity [52,84,85,86]. Clinically, GCAs exhibit an intermediate behavior between neuroendocrine neoplasms (NENs) and conventional adenocarcinomas. Average 5-year survival has been reported at approximately 72%, with one- and two-year survival estimates of 92.4% and 88.8%, respectively [43,68,87,88,89,90,91,92,93] (Figure 2b).

Genomic studies suggest that GCA harbors a mutational profile distinct from other appendiceal neoplasms. Specifically, *KRAS* (6.9%, range: 0–8.3%), *GNAS* (2.9%, 1–3.8%), and *APC* (3.8%, 0–11.8%) mutations are far less frequent in GCAs than in MAAs or CTAs [12,35,36,37,38,39,52,55,94,95,96] (Figure 3c). Instead, recurrent mutations have been observed in *ARID1A* (11.8%), *ARID2* (11.8%), *CDH1* (11.8%), *RHPN2* (11.8%), and *MLL2*/*KMT2D* (8.8%) [37] (Table 1). Interestingly, the *RHPN2* gene, which encodes a Rho GTPase-binding protein, has been implicated in promoting epithelial-to-mesenchymal transition (EMT) in gliomas through activation of RhoA signaling [97,98], suggesting a similar functional consequence of *RHPN2* mutation in GCAs.

Chromatin remodeling has also emerged as a potential pathogenic mechanism in GCAs. In one study, 4 of 9 cases harbored truncating or splice site mutations in *ARID1A*, *KDM6A*, or *KMT2D*—all major regulators of chromatin structure and gene expression [36]. Additional alterations in *SOX9* (frameshift), *ERBB3* (missense), and *EDNRB* (nonsense) were also reported in the same cohort. Loss of heterozygosity in chromosomes 11q, 16q, and 18q have been reported in up to 25%, 38%, and 56% of GCAs, suggesting that the loss of these chromosomes may drive GCA pathogenesis [99].

Clonality studies comparing “pure” goblet cell carcinoids and adenocarcinoma ex-goblet cell carcinoids (GCAs that are morphologically distorted or present disbursed goblet cell clusters) revealed shared *TCF7L2* mutations, implicating Wnt signaling and confirming a common origin [37,100]. Of note, *BRAF* mutations, which are frequently observed in colorectal and other gastrointestinal cancers, appear to be absent in GCAs [12,92], consistent with a divergence from canonical *RAS/RAF*-driven tumorigenesis.

## 5. Appendiceal Colonic-Type Adenocarcinoma

Colonic-type adenocarcinomas (CTAs), also known as intestinal-type adenocarcinomas, are non-mucinous adenocarcinomas that share mutational and clinical features with colorectal adenocarcinomas [66]. CTAs frequently arise at the base of the appendix and are often diagnosed incidentally during surgical treatment for acute appendicitis [65]. SEER survival data suggest that CTAs have a 1-year survival rate of 83%, a 2-year survival rate of 61%, and a 5-year survival rate of approximately 61% [3,6,12,25,44,67,68,69,77,101,102] (Figure 2b). Historically, CTAs were thought to represent a more aggressive subtype compared to mucinous adenocarcinomas (MAAs). However, a retrospective SEER analysis from 1973 to 1988 reported no statistically significant difference in survival between mucinous and non-mucinous appendiceal adenocarcinomas [68,69].

The mutational spectrum of CTAs largely overlaps with that of MAAs, particularly in genes involved in RAS/MAPK signaling and tumor suppressor pathways. Common mutations include *KRAS* (average: 54.8%, range: 36.4–75%), *TP53* (43.8%, 18.2–71.4%), and *GNAS* (26%, 13.5–31%) [12,32,38,39,40,66,103] (Figure 3d). GNAS mutations appear more frequently in late-onset than in early-onset disease [76], though the clinical implications of this distinction remain to be elucidated.

Mutations in the RAS/RAF signaling pathway, including *BRAF*, *HRAS*, *KRAS*, and *NRAS*, are observed in up to 60% of CTAs [12]. These findings highlight the central role of MAPK/ERK signaling in CTA tumorigenesis. By contrast, alterations in the *PI3K/AKT* pathway are comparatively rare in CTAs and MAAs (~9% and 8%, respectively), suggesting a relatively minor contribution of this pathway to CTA pathobiology [3,9,32,35,38,39,40,70,95]. Other commonly mutated genes in CTAs include *SMAD4* and APC, indicating potential involvement of the TGF-β and Wnt signaling pathways, respectively [12,39].

CTA microsatellite instability rates have been reported to more closely match that of CRC, with up to 18% of CTAs presenting with high MSI. MSI-high status may be dependent on CTA differentiation, as half of poorly differentiated CTAs, and no well- or moderately differentiated CTAs, were MSI-high [80]. Similar to CRC, the loss of chromosome 18q—a region harboring tumor suppressors *SMAD2*, *SMAD4*, and *DCC*—is frequently observed in CTAs and has been associated with poor prognosis in colorectal cancers [104,105]. In one study, 5 of 8 CTAs and 7 of 13 MAAs exhibited 18q deletions, implicating this event in the pathogenesis of both subtypes [104].

## 6. Appendiceal Signet Ring Cell Adenocarcinoma

Signet ring cell adenocarcinoma (SRC) is the rarest and most lethal histologic subtype of appendiceal cancer, which comprises approximately 2% of all AC cases [5,68,89] (Figure 2a). SRC is characterized by its distinctive cell morphology: poorly cohesive malignant cells with intracellular mucin displacing the nucleus, yielding a “signet ring” appearance [106]. True SRCs exhibit a diffuse cord-like infiltrative growth pattern with minimal gland formation and often overlap histologically with poorly differentiated mucinous adenocarcinomas [6,19,107]. SRCs carry the worst prognosis among appendiceal cancers. SEER-based survival analyses have demonstrated a 1-year survival rate of 85%, a 2-year rate of 66%, and markedly diminished 5-year survival rate of 21% [3,6,12,24,67,68,69,77,101,102] (Figure 2b).

Given the rare and deadly nature of SRC, molecular and genomic characterization studies have remained limited [7,107]. As extrapolated from multiple reports, the most frequently mutated genes are *TP53* (average: 34.2%, range = 15.4–43%) and *SMAD4* (20.5%, range = 11–30%), both with mutation rates similar to that observed in CTAs [12,32,39,40,108] (Figure 3e). Unlike MAAs or CTAs, SRCs exhibit lower frequencies of *KRAS* mutation (average: 19%, range = 7–35%) [12,32,39,40]. This low occurrence of RAS pathway activation may reflect a divergent oncogenic mechanism in SRC as compared to other adenocarcinomas of the appendix. Other mutations frequently observed in high-grade appendiceal tumors, such as those affecting *APC* or *PIK3CA*, have not been consistently reported in SRCs, which may owe to the low incidence of this subtype and limited genomic sampling. In a small study of twenty-three appendiceal SRC cases, none presented high MSI [80].

## 7. Molecular Divergence Between Appendiceal and Colorectal Cancers

While AC and CRC share anatomic origins within the gastrointestinal tract, accumulating evidence suggests that they are molecularly distinct entities. This distinction can be observed in the differential mutational profiles of several canonical oncogenic driver genes, particularly those within the *TP53*, *APC*, *KRAS*, *GNAS*, *SMAD4*, *BRAF*, and *PIK3CA* signaling axes [109,110]. A 2020 report comparing 193 appendiceal adenocarcinoma (AA) patients to 2860 CRC patients [9] identified comparable rates of mutation in SMAD4 (16.9% vs. 13.6%), with AA presenting higher rates of GNAS (28% vs. 2%) mutation and lower rates of TP53 (27% vs. 68%) and APC (9% vs. 55%) compared to CRC. Notably, the authors reported that AA ctDNA sensitivity was significantly lower in AA (particularly low grade AA) relative to CRC (e.g., reduced detection of *TP53* variants), suggesting ctDNA’s clinical utility may be limited to only high-grade AA patients.

In CRC, mutations in *TP53* and *APC* are hallmark features of the disease that occur at frequencies approaching 70% for both genes [107,111,112]. However, these genes are mutated at significantly lower frequencies across most subtypes of AC. In mucinous adenocarcinomas (MAAs), *TP53* mutations are detected in approximately 28% of cases overall, with enrichment in high-grade tumors (47%) relative to low-grade tumors (32%) [72,74]. Similarly, *APC* mutations are relatively infrequent across AC subtypes, with reported rates of 12.2% in ANENs [30,35,40], 3.8% in goblet cell adenocarcinoma (GCA) [12,35,37,38,39,52,77], 8.0% in MAAs [9,12,32,34,35,38,39,40,71], and 6.4% of SRCs [12,32,40]. CTAs demonstrate moderately higher *APC* mutation rates with an average mutational frequency of 15.5% [12,32,36,38,40,95]. In sum, these rates fall short of the rates observed in CRC, consistent with a fundamental divergence in oncogenic Wnt signaling between the two tumor types [9,113].

Another point of divergence lies in the *GNAS* gene, which is rarely mutated in CRC [111] but is among the most frequently altered genes in appendiceal mucinous tumors. Mutations in *GNAS* occur in ~44% of MAAs overall [72] and are particularly prevalent in low-grade disease (up to 70%) [3,12,34,75]. Known oncogenic driving mutations of *GNAS* include R201C/H gain-of-function mutations, which have been implicated in the oncogenesis of gastrointestinal and pancreatic tumors [113,114]. This contrast suggests a prominent role for the Gsα-cAMP-PKA and mucin signaling pathways in appendiceal tumorigenesis that is not shared with CRC.

*KRAS* mutations, on the other hand, are observed in both appendiceal and colorectal tumors, but with a more nuanced profile. In CRC, *KRAS* mutations are present in 40–50% of cases [111,115], while in appendiceal tumors, *KRAS* mutation rates are comparable or even higher in some subtypes. *KRAS* mutation rates have been observed in up to 67% of MAAs [9,12] and 55% of CTAs [12,32] but are notably infrequent in GCAs (~7%) [12] and SRCs (~19%) [12]. While oncogenic *KRAS* signaling has a role in the pathobiology of both cancer types, subtype-specific differences underscore the molecular heterogeneity of AC.

*SMAD4*, a tumor suppressor in the TGF-β pathway, is altered in approximately 14% of CRCs [111,115]. In AC, *SMAD4* mutations are reported at similar or higher frequencies in certain high-grade subtypes, including 20.5% in SRC [12,32,39,40] and 24% in high-grade MAAs [79]. However, this mutation is rarely observed in low-grade AC, suggesting a functional association with more aggressive cancers.

Mutations in *PIK3CA*, which occur in 18–20% of CRCs [13] are comparatively infrequent in AC, with reported rates of 17% in MAAs [3,32] and even lower in CTAs and other subtypes [38,39]. Similarly, *BRAF* mutations, which are present in approximately 10% of CRCs and associated with poor prognosis, are rarely observed in AC tumors at rates as low as 3% [116]. Despite the rarity of *BRAF* mutations in AC, *BRAF V600E*-targeted inhibition has demonstrated disease control rates of 80% [116]. Microsatellite instability (MSI) is observed in approximately 15% of CRCs [107,112,117] and is often used as a biomarker for immunotherapy response. However, MSI is exceedingly rare in AC and has been documented in less than 1% of MAAs [80,96], suggesting fundamental differences in DNA repair activity between the two cancer types.

Transcriptomic variation between AA and CRC has also been observed. In a microarray profiling study of AC (*n* = 26) and CRC (*n* = 15), unsupervised hierarchical cluster analysis demonstrated that global gene expression could significantly differentiate AA and CRC cases [118]. Three principal tumor clusters were identified, two of which were comprised predominantly of AA cases (with several CRC tumors co-clustering with AA), and the other comprised mostly of CRC cases (with one AA tumor co-clustering with CRC). The co-clustering samples were not studied further, but could reflect rare convergent transcription events related to tumor heterogeneity. In addition, the three tumor clusters were shown to vary by survival, with the CRC-predominant cluster exhibiting the worst prognosis.

Together, these data reinforce the notion that AC, despite sharing a gross anatomical origin with CRC, arises from distinct molecular pathways of tumorigenesis. The low prevalence of *APC* and *TP53* mutations, the rarity of *BRAF* alterations, and the predominance of *GNAS* mutations together argue against the classification of AC as an anatomical variant of CRC. Rather, AC should be viewed as a biologically distinct cancer with subtype-specific genomic alterations that require further investigation to help inform the development of tailored therapeutic strategies.

## 8. Mutated Genes with Potential Clinical Actionability in AC

Identifying clinically actionable mutations in AC remains a challenge given its rarity and histologic heterogeneity, plus the limited availability of genome-wide sequencing data [118,119,120,121,122,123]. However, a number of genetic alterations have emerged as potential therapeutic targets in AC.

*KRAS* mutations are among the most common across AC subtypes and represent a promising yet difficult target. *KRAS* is a well-established oncogene in many solid tumors, but the development of inhibitors has been hampered by the protein’s structure, mainly due to a lack of accessible binding sites on RAS-associated proteins [124]. Drugs specifically designed to target the *KRAS* G12C mutation, such as Sotorasib and Adagrasib, inhibit oncogenic *KRAS* signaling and have demonstrated response rates of 33–43% in metastatic CRC [125]. Analysis of the Foote et al. data set (via cBioPortal: https://www.cbioportal.org/study/summary?id=appendiceal_msk_2022 accessed on 9 September 2025) identified *KRAS* G12C mutations in 6.1% of MAAs (*n* = 10/164), suggesting a small but potentially targetable patient population [38]. KRAS G12C mutations are of clinical interest because this mutation is targeted by clinically efficacious inhibitors such as Sotorasib, which holds promise for this minority of AC patients. No G12C mutations were found in GCAs (*n* = 0/72) or CTAs (*n* = 0/37). For tumors harboring these and other KRAS mutations, multiple targeted approaches are in development, including KRAS G12D inhibitors (e.g., Zoldonrasib) and pan-KRAS inhibitors (e.g., Daraxonrasib), which have shown promising results in pre-clinical models and early clinical trials [126,127].

Currently, for other *KRAS* variants, targeting downstream of *KRAS* in the MEK-ERK or PI3K/AKT pathways may prove more efficacious [34,78]. An analysis of 78 appendiceal colonic type adenocarcinomas found that ~5% of cases exhibited one or more potentially druggable targetable mutations in *ALK*, *EGFR*, *MET*, *IDH1* and *ERBB2*—all of which intersect the RAS/RAF signaling cascade [71].

Another emerging target is *PIK3CA*, which encodes the catalytic subunit of PI3K. Although *PIK3CA* mutations are relatively infrequent in AC, FDA-approved inhibitors such as alpelisib, which has been used successfully in *PIK3CA*-mutated breast cancer, may be an effective treatment approach for a subset of AC patients, such as MAAs [128,129].

## 9. New Technologies for Understanding AC Pathophysiology and Therapeutic Response

Recent advances in single-cell profiling are beginning to illuminate the biology of appendiceal cancer (AC), though relevant models remain limited [12,25]. In a recent study comparing low- and high-grade mucinous neoplasms (MANs) with matched normal tissue, defining goblet cell-like features predominated adenocarcinomas not otherwise pathologically defined as GCAs, suggesting a pervasive goblet cell, or goblet cell-precursor lineage among MANs [130]. Prospective analyses of the intersection between AC histology and cellular-genomic architecture using modern spatial single-cell transcriptomics and epigenomic technologies will further clarify cell-of-origin, and link cellular organization and pathobiology to patient outcomes.

Modeling AC remains challenging due to the low cellularity of mucin-rich tumors and the absence of a defined precancerous lesion [12,24]. Patient-derived tumor organoids (PTOs) have emerged as promising pre-clinical models that are robust to low cellularity and the mucinous phenotype, enabling ex vivo modeling of AC pathophysiology and drug responsiveness, and demonstrating prognostic power for predicting patient clinical benefit [129,131]. Notably, comparable predictive accuracy has also been reported in colorectal cancer PTOs [132,133], underscoring their utility across gastrointestinal tumor types. Through integration of PTO models with genomic and transcriptomic profiling, researchers may be able to identify patient subsets predictive of a range of therapeutic and surgical outcomes, including those most likely to benefit from cytoreductive surgery and/or HIPEC [21,122,123,129,131]. The establishment of advanced pre-clinical models represents a critical step toward precision oncology in AC, offering functional evidence to guide treatment selection in a setting where therapeutic options remain empiric and outcomes variable.

## 10. Discussion

A comprehensive understanding of the genetic underpinnings of appendiceal cancer across its histological subtypes is essential for improving diagnostic precision, prognostication, and therapeutic decision-making. This review highlights key mutational differences across the five major histologic subtypes of AC, which provides new insights into their heterogeneity, relevance in disease progression, and potential clinical actionability.

*KRAS* and *GNAS* mutations are characteristic of low-grade appendiceal cancers and are generally associated with more indolent clinical behavior [12,25,77,78]. By contrast, high-grade subtypes such as CTAs, SRCs and high-grade MAAs exhibit frequent mutations in *TP53*, *APC*, and *SMAD* gene family members, reflecting their involvement in more aggressive AC tumor phenotypes.

Among AC subtypes, *TP53* mutations are most prevalent in CTAs (~44%) and SRCs (~34%), followed by MAAs (~28%), with the lowest frequency observed in neuroendocrine neoplasms (NENs, ~23%) and goblet cell adenocarcinomas (GCAs, ~21%). Similarly, *APC* mutations are enriched in high-grade CTAs, SRCs, and MAAs—reminiscent of the mutational spectrum of colorectal cancer and indicative of a shared dependency on Wnt pathway activation in these more lethal subtypes [25,134]. Components of the TGF-β signaling pathway, *SMAD2*, *SMAD3* and *SMAD4*, are also frequently altered in high-grade MAAs thereby linking this pathway to MAA disease progression and metastatic potential.

At present, the strongest independent predictors of survival outcome in AC remain the traditional clinicopathologic factors: histologic subtype, tumor grade, extent of disease at diagnosis, performance status, and completeness of surgical cytoreduction. The attendant comorbidity associated with CRS/HIPEC for the management of disseminated AC tumors emphasizes the value of prognostic genetic signatures in the diagnostic setting, particularly for the delineation of HIPEC eligibility [122,123]. An example where genomic insights can provide practice-guiding information is in AC patients that would be traditionally excluded from receiving post-operative HIPEC due to overwhelming tumor burden in the peritoneal cavity prior to cytoreductive surgery. Despite clinical experience associating tumor burden with refractory disease, pre-operative genomic profiling can shed insight on the molecular programming associated with favorable HIPEC response independent of pre-operative tumor burden, in turn increasing the eligibility pool to receive curative CRS/HIPEC rather than palliative systemic chemotherapy.

Building on these molecular insights, recent work by Foote et al. [38] showed that co-occurring mutations in RAS, *GNAS*, and *TP53* could be used to classify appendiceal adenocarcinomas into molecular subtypes with distinct clinical trajectories. Specifically, RAS-mutant/*GNAS*-wildtype/*TP53*-wildtype tumors defined a genomically quiet, clinically indolent subtype with minimal stromal invasion and exceptional survival outcomes that persisted in the metastatic setting. Tumor clonality analyses revealed that RAS mutations tend to occur early in tumor evolution, preceding the acquisition of *TP53* or *GNAS* mutations. This temporal ordering suggests that RAS-mutant tumors are molecularly young, a feature that may underlie their favorable prognosis. In contrast, *TP53*-mutant tumors exhibited high aneuploidy, destructive stromal invasion and poor survival, while *GNAS*-mutant tumors showed chemoresistance and elevated peritoneal tumor burden. Notably, these molecular subtypes retained prognostic significance independent of histologic grade and subtype, thus demonstrating the clinical utility of integrated genomic profiling in AC.

In current practice, diagnostic targeted gene sequencing panels offer the convenience of cheap cost production at scale, and have enabled the relatively quick turnaround on AC genomic assessment. However, the comprehensive molecular characterization of appendiceal cancers, at the level needed to reveal meaningful new insights, will require greater oncogenic granularity such as that afforded by whole-exome and full-genome sequencing strategies. Another limitation, and one that no doubt impacts the accuracy of the data synthesized in this review, is study-to-study variation in sample collection procedures, quality control practices, sequencing methods used, and cohort size—all factors that contribute to inherent study bias. For the effective characterization of AC and its histologic variants, standard clinical and analytical protocols must be developed and implemented.

As molecular insights into appendiceal cancer continue to evolve, these findings collectively support a shift beyond histopathologic classification and toward a molecularly informed framework for appendiceal cancer—one that enhances our understanding of AC biology and directs mechanistic studies to elucidate how recurrent mutations shape tumor evolution, clinical behavior, and therapeutic vulnerability.

## 11. Conclusions

Preliminary efforts to implement gene-panel profiling in AC have yielded promising results [118,123,135], but large-scale, subtype-specific genomic studies remain critically needed. The prioritization of deep sequencing technologies, such as whole-exome sequencing, whole-genome sequencing and single-cell analysis, will be paramount for prospective AC research. Such approaches will shed light not only on the full mutational spectrum, but also on the chromosomal and copy number changes that underlie the etiology of AC and allow new insight into the AC tumor microenvironment. Equally important will be the development of physiologically relevant models, including patient-derived organoids [136] and xenografts [137] that can recapitulate AC biology and support mechanistic studies. Ultimately, through the delineation of subtype-specific molecular drivers, the field is poised to usher in a new era of precision oncology for appendiceal cancer, where therapeutic interventions are tailored to the unique genetic profiles of individual tumors.

## Figures and Tables

**Figure 1 cancers-17-03275-f001:**
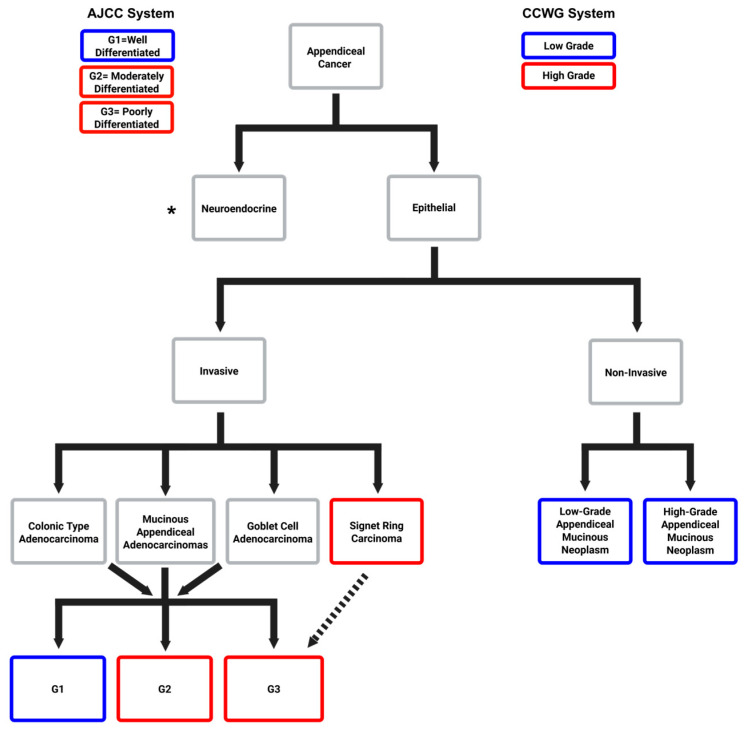
Consolidation of appendiceal cancer grading schemas between the 8th Edition AJCC Guidelines and Chicago Consensus Working Group (CCWG). Solid arrows indicate primary pathological subclassifications. Invasive histologic variants (i.e., Neuroendocrine, Colonic Type, Goblet Cell Adenocarcinoma) can be further classified into grades G1–G3. The dashed arrow indicates a secondary subclassification with regard to grade: any histologic variant with a signet ring component is clinically assessed as high-grade or G3 disease. * Appendiceal neuroendocrine neoplasms (ANENs) are typically non-aggressive and associate with favorable survival; however, poorly differentiated ANENs are considered malignant and high-grade (Made with bioRender).

**Figure 2 cancers-17-03275-f002:**
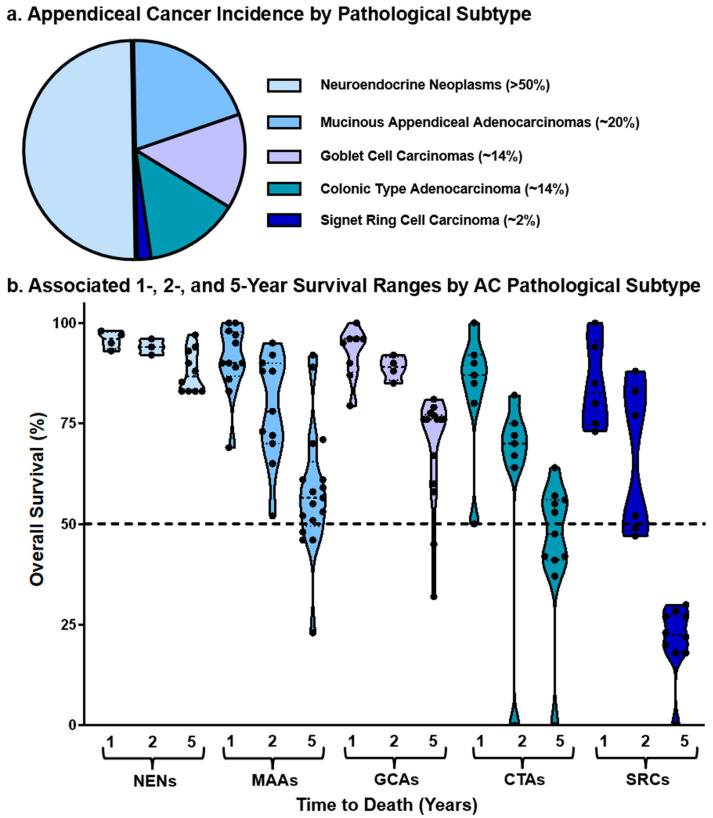
(**a**), Incidence of appendiceal cancer subtypes and (**b**), mortality by pathological subtype. NEN = Neuroendocrine Neoplasm. GCA = Goblet Cell Adenocarcinoma. MAN = Mucinous Appendiceal Neoplasms. CTA = Colonic Type Adenocarcinoma. SRC = Signet Ring Cell Adenocarcinoma. Reference data for (**a**,**b**) can be found in Appendix A. Each data point in (**b**) represents a reported survival rate as listed from an individual study.

**Figure 3 cancers-17-03275-f003:**
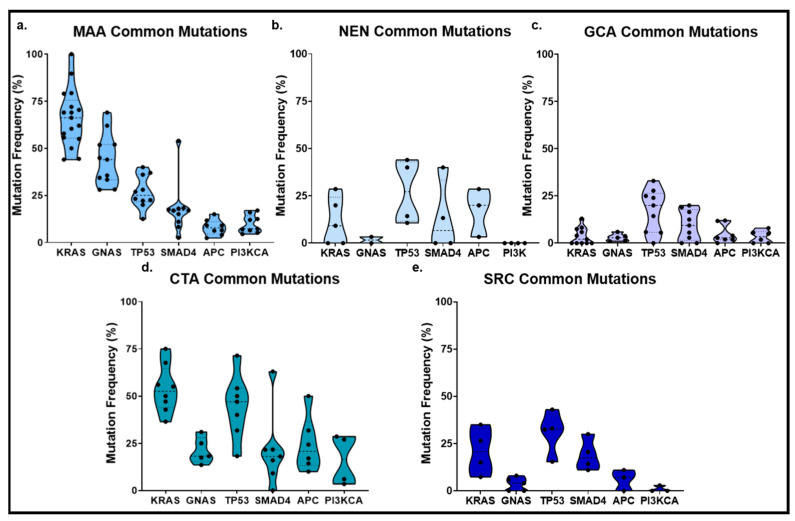
Frequencies of common oncogenic mutations by histologic subtype. Shown are mutation frequencies extrapolated from multiple reports. (**a**), Neuroendocrine Neoplasms (NEN). (**b**), Mucinous Appendiceal Neoplasms (MAN). (**c**), Goblet Cell Adenocarcinoma (GCA). (**d**), Colonic Type Adenocarcinoma (CTA). (**e**), Signet Ring Cell Adenocarcinoma (SRC). Reference data for (**a**–**e**) can be found in Appendix A. Each data point in (**a**–**e**) represents a reported gene mutation rate as listed from an individual study.

**Table 1 cancers-17-03275-t001:** Additional gene alterations observed in AC subtypes.

Altered Gene	AC Subtype	Percent of Cases per Study (No Pts) in Reference Order	Reference(s)
EGFR	NEN	40% (5)	[30]
RN43	MAA	44% (9), 7% (44), 33% (9)	[31,32,33]
SMAD2	MAA	26% (29)	[34]
SMAD3	MAA	10% (10)	[34]
TGFBR1	MAA	10% (10)	[34]
TGFBR2	MAA	10% (10)	[34]
ARID1A	GCA	15% (84), 15.4% (13), 33% (9), 11% (18)	[12,35,36,37]
FAT3	GCA	17% (84)	[12]
MSH2	GCA	17% (18)	[37]
SOX9	GCA	22% (9), 10% (72), 9.2% (141), 11% (18)	[36,37,38,39]
ARID1A	CTA	11% (208)	[12]
ARID1A	SRC	7% (37), 11% (27)	[12,32]
CDH1	SRC	7.1% (14)	[40]
LRP1B	SRC	16.7% (24)	[39]
PRKDC	SRC	12.0% (50)	[39]

## Data Availability

This is a systematic review; no new data were created.

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
