# Peer review of "The Genomic Topography of Appendiceal Cancers: Our Current Understanding, Clinical Perspectives, and Future Directions"

_cancers, 2025, doi:10.3390/cancers17193275_

Round 1

Reviewer 1 Report

Comments and Suggestions for Authors

Here Gironda et al review genomic characterization of appendix cancer. Overall I support

publication. Below are some comments / suggestions.

Figure 1 – overall a nice figure, however Signet Ring Adenocarcinoma is a distinct subtype of AA

and should be on the same line as colonic type, mucinous, and GCA; with a line connecting to G3

(all SRC are grade 3).

What is the source of the data in Fig 2? For Fig 2B and Fig 3 what do each point represent?

“KRAS mutations were present 184 in 61% of G1, 72% of G2, and only 19% of G3 tumors [25].”

-although reported in that reference, should cite other references that report KRAS mutation is

frequent in G1, G2 and G3 tumors.

“However, this mutation is rarely observed in low-grade AC, indicating a potential role in disease

progression rather than cancer initiation.”

-this suggests that the authors feel that high grade tumor progress from low-grade, which is not

generally the consensus. This should be changed or clarified

Line 304 – in this section contrasting AA and CRC, would reference (Raghav, Shen et al. 2020) and

add 1-2 sentences re transcriptional differences between CRC and AA. Would consider also 1-2

sentences on ctDNA which is another contrast between AA and CRC.

Line 321 – when discussing role of GNAS consider referencing paper demonstrating its

oncogenesis (More, Ito et al. 2022)

Line 341 – when discussing BRAF mutation in AA would ref (Pattalachinti, Haque et al. 2025)

Lines 359-368 – should mention KRAS G12D inhibitors and pan-KRAS inhibitors

Table 2 – anti-EGFR antibodies are ineffective in KRAS mutant tumors, this makes me question the

entire DGIdb system. I would favor removing table 2. Without any clinical or preclinical data

suggesting efficacy of these drugs these data seem to have little value.

Gene names should be italicized

Author Response

  1. Figure 1 – overall a nice figure, however Signet Ring Adenocarcinoma is a distinct subtype of AA and should be on the same line as colonic type, mucinous, and GCA; with a line connecting to G3 (all SRC are grade 3).”

We have amended Figure 1 to show SRC as a distinct subtype of AA, with a connecting dotted line to emphasize that any subtype of AC with a SRC component is graded G3.

In the Figure 1 legend, we added “Any histologic variant containing a signet ring component is clinically assessed as high-grade or G3 disease.”

  1. “What is the source of the data in Fig 2? For Fig 2B and Fig 3 what do each point represent?”

We have included our Excel data file that tracks this information as Supplementary Data 1.

To address the source of data of Figure 2 and Figure 3 in the manuscript itself, we have included the following statements in the figure legends:

  1. “Reference data for Figure 2a,b can be found in Supplementary Data 1. Each data point in figure 2b represents a reported survival rate as listed from an individual study.”

  1. “Reference data for Figure 3 can be found in Supplementary Data 1. Each data point in figure 3a-3e represents a reported gene mutation rate as listed from an individual study.”

  1. KRAS mutations were present 184 in 61% of G1, 72% of G2, and only 19% of G3 tumors [25].” -although reported in that reference, should cite other references that report KRAS mutation is frequent in G1, G2 and G3 tumors.”.

We added the following sentence to address this point:

“Analysis of open-source data on 164 MAA tumors generated by an MSK-Impact Trial that followed the WHO classification system [70] showed a similar grade-dependent trend with KRAS mutations rates of 88% (G1), 83% (G2), and 53% (G3) – the higher overall frequencies possibly owing to variation in grade classification system or sequencing method.”

  1. “However, this mutation is rarely observed in low-grade AC, indicating a potential role in disease progression rather than cancer initiation.” -this suggests that the authors feel that high grade tumor progress from low-grade, which is not generally the consensus. This should be changed or clarified”

We have amended the sentence to read: “However, this mutation is rarely observed in low-grade AC, suggesting a functional association with more aggressive cancers.”

  1. Line 304 – in this section contrasting AA and CRC, would reference (Raghav, Shen et al. 2020) and add 1-2 sentences re transcriptional differences between CRC and AA. Would consider also 1-2 sentences on ctDNA which is another contrast between AA and CRC.

  • We have included the following sentences to highlight the information in Raghav et al 2020:

“A 2020 report comparing 193 appendiceal adenocarcinoma (AA) patients to 2,860 CRC patients [9] identified comparable rates of mutation in SMAD4 (16.9% vs 13.6%), with AA presenting higher rates of GNAS (28% vs 2%) mutation and lower rates of TP53 (27% vs 68%) and APC (9% vs 55%) compared to CRC. Notably, the authors reported that AA ctDNA sensitivity was significantly lower in AA (particularly low grade AA) relative to CRC (e.g., reduced detection of TP53 variants) suggesting ctDNA’s clinical utility may be limited to only high-grade AA patients.”

  • We have included the following sentences to discuss the transcriptional differences between AC and CRC:

“Transcriptomic variation between AA and CRC has also been observed. In a microarray profiling study of AC (n=26) and CRC (n=15), unsupervised hierarchical cluster analysis demonstrated that global gene expression could significantly differentiate AA and CRC cases [119]. Three principal tumor clusters were identified, two of which were comprised predominantly of AA cases (with several CRC tumors co-clustering with AA), and the other comprised mostly of CRC cases (with one AA tumor co-clustering with CRC). The co-clustering samples were not studied further, but could reflect rare convergent transcription events related to tumor heterogeneity. In addition, the three tumor clusters were shown to vary by survival, with the CRC-predominant cluster exhibiting the worst prognosis.” 

  1. “Line 321 – when discussing role of GNAS consider referencing paper demonstrating its oncogenesis (More, Ito et al. 2022)”

We have included the More et al citation with the following sentence: “…Known oncogenic driving mutations of GNAS include R201C/H gain-of-function mutations, which have been implicated in the oncogenesis of gastrointestinal and pancreatic tumors [131].”

  1. “Line 341 – when discussing BRAF mutation in AA would ref (Pattalachinti, Haque et al. 2025)”

We have included the following sentence to discuss the clinical significance of BRAF mutation in AC: “Despite the rarity of BRAF mutations in AC, BRAF V600E-targeted inhibition has demonstrated disease control rates of 80% [132].”

  1. “Lines 359-368 – should mention KRAS G12D inhibitors and pan-KRAS inhibitors”

The following sentence has been included to address this comment: “KRAS G12C mutations are of clinical interest because this mutation is targeted by clinically-efficacious inhibitors such as Sotorasib, which holds promise for this minority of AC patients. No G12C mutations were found in GCAs (n=0/72) or CTAs (n=0/37). For tumors harboring these and other KRAS mutations, multiple targeted approaches are in development, including KRAS G12D inhibitors (e.g., Zoldonrasib) and pan-KRAS inhibitors (e.g., Daraxonrasib), which have shown promising results in pre-clinical models and early clinical trials [137,138].

  1. “Table 2 – anti-EGFR antibodies are ineffective in KRAS mutant tumors, this makes me question the entire DGIdb system. I would favor removing table 2. Without any clinical or preclinical data suggesting efficacy of these drugs these data seem to have little value

Table 2 has been omitted as suggested.

  1. “Gene names should be italicized

We have italicized all gene names as suggested.

Reviewer 2 Report

Comments and Suggestions for Authors

The manuscript is promising and of potential high impact. However, revisions are necessary to strengthen methodological transparency and refine the translational perspective to emphasize clinical utility.

  1. The review primarily summarizes reported frequencies but offers limited critique of the heterogeneity in sequencing methods, sample sizes, and study biases. A more explicit evaluation of methodological limitations across studies (e.g., targeted panels vs. whole-exome sequencing) would strengthen the conclusions.
  2. While Table 2 lists FDA-approved drugs targeting certain genes, the discussion of actual feasibility in AC is limited. More caution is needed in distinguishing theoretical versus validated targets.
  3. The review would benefit from a dedicated section on how novel approaches (single-cell sequencing, spatial transcriptomics, organoid models) can specifically advance AC research, rather than brief mentions.
  4. The manuscript could strengthen its translational impact by addressing how genomic insights might inform current clinical management pathways (e.g., HIPEC eligibility, stratification of adjuvant therapies).

Author Response

  1. “The review primarily summarizes reported frequencies but offers limited critique of the heterogeneity in sequencing methods, sample sizes, and study biases. A more explicit evaluation of methodological limitations across studies (e.g., targeted panels vs. whole-exome sequencing) would strengthen the conclusions.”

We have included the following language in the introduction and conclusions, respectively:

  1. “Most studies to date have been derived from small cohorts and targeted sequencing efforts using a variety of diverse gene panels including the MSK-IMPACT, Ion Ampliseq Cancer Panel, Oncomine Comprehensive Cancer Panel, and Ion AmpliSeq Comprehensive Can-cer panel, which vary widely with respect to the breadth and type of genes represented.”

  1. “In current practice, diagnostic targeted gene sequencing panels offer the convenience of cheap cost production at scale, and have enabled the relatively quick turnaround on AC genomic assessment. However, the comprehensive molecular characterization of appendiceal cancers, at the level needed to reveal meaningful new insights, will require greater oncogenic granularity such as that afforded by whole-exome and full-genome sequencing strategies. Another limitation, and one that no doubt impacts the accuracy of the data synthesized in this review, is study-to-study variation in sample collection procedures, quality control practices, sequencing methods used, and cohort size – all factors that contribute to inherent study bias. For the effective characterization of AC and its histologic variants, standard clinical and analytical protocols must be developed and implemented. ”

  1. “While Table 2 lists FDA-approved drugs targeting certain genes, the discussion of actual feasibility in AC is limited. More caution is needed in distinguishing theoretical versus validated targets.

Based on the recommendation of other reviewers, Table 2 has been omitted from the manuscript.

  1. “The review would benefit from a dedicated section on how novel approaches (single-cell sequencing, spatial transcriptomics, organoid models) can specifically advance AC research, rather than brief mentions… “The manuscript could strengthen its translational impact by addressing how genomic insights might inform current clinical management pathways (e.g., HIPEC eligibility, stratification of adjuvant therapies)”

We have included a new section (9) titled “New Technologies for Understanding AC Pathophysiology and Therapeutic Response”. The following text has been included within the section:

“Recent advances in single-cell profiling are beginning to illuminate the biology of appendiceal cancer (AC), though relevant models remain limited [12,25]. In a recent study com-paring low- and high-grade mucinous neoplasms (MANs) with matched normal tissue, defining goblet cell-like features predominated adenocarcinomas not otherwise pathologically defined as GCAs, suggesting a pervasive goblet cell, or goblet cell-precursor lineage among MANs [136]. Prospective analyses of the intersection between AC histology and cellular-genomic architecture using modern spatial single-cell transcriptomics and epigenomic technologies will further clarify cell-of-origin, and link cellular organization and pathobiology to patient outcomes.

Modeling AC remains challenging due to the low cellularity of mucin-rich tumors and the absence of a defined precancerous lesion [12,24,124,133,136]. Patient-derived tu-mor organoids (PTOs) have emerged as promising pre-clinical models that are robust to low cellularity and the mucinous phenotype, enabling ex vivo modeling of AC pathophysiology and drug responsiveness, and demonstrating prognostic power for predicting patient clinical benefit [124,133]. Notably, comparable predictive accuracy has also been reported in colorectal cancer PTOs [134,135], underscoring their utility across gastrointestinal tumor types. Through integration of PTO models with genomic and transcriptomic profiling, researchers may be able to identify patient subsets predictive of a range of therapeutic and surgical outcomes, including those most likely to benefit from cytoreductive surgery and/or HIPEC [21,119,120,124,133]. The establishment of advanced pre-clinical models represents a critical step toward precision oncology in AC, offering functional evidence to guide treatment selection in a setting where therapeutic options remain empiric and outcomes variable.”